# Healthy Urban Environmental Features for Poverty Resilience: The Case of Detroit, USA

**DOI:** 10.3390/ijerph18136982

**Published:** 2021-06-29

**Authors:** Patricia Leandro-Reguillo, Amy L. Stuart

**Affiliations:** 1Sensibilicity, Heidelberg 69117, Germany; 2College of Public Health, University of South Florida, Tampa, FL 33617, USA; als@usf.edu

**Keywords:** poverty resilience, urban health, urban environment, accessibility, healthy lifestyle

## Abstract

Within the existing relationship among urban environment, health, and poverty, it is necessary to clarify and characterize the influence that the physical environment has on community socioeconomic outcomes. Given that Detroit has one of the highest poverty rates among large metropolitan areas in the United States, this study aims to identify environmental and urban features that have influenced poverty in this city by assessing whether changes in household income are associated with characteristics of the built environment. The difference of median household income (DMHI) between 2017 and 2013 and 27 environmental and urban variables were investigated using both geographic distribution mapping and statistical correlation analysis. Results suggest that proximity of housing to job opportunity areas, as well as to certain educational and health-related facilities, were positively related to increasing household incomes. These findings outline a healthy urban design that may benefit community socioeconomic outcomes—specifically a design with dense and mixed-use areas, good accessibility, high presence of urban facilities, and features that promote a healthy lifestyle (involving physical activity and a healthy diet). In this sense, urban planning and public health may be important allies for poverty resilience.

## 1. Introduction

During the first decade of this century, the United States experienced a dramatic increase in the number of people living in extreme poverty. Although public policies in the 1990s reduced the number of high-poverty neighborhoods by over 40 percent, the number of people living in high-poverty areas rose from 7.2 million in 2000 to 13.8 million in 2013 [1]. This increase in poverty was characterized by a pattern of higher concentration and segregation in large- to medium-sized metropolitan areas, with high persistence of extreme poverty [1]. Increased poverty is a substantial concern because it directly affects public health. The poor have higher levels of disease, higher child and maternal mortality, and worse access to health care; they also die younger [2].

Economic, social, political, and health determinants have historically been assumed to be primarily responsible for poverty occurrence and dynamics [2,3]. Characterizing other factors that may influence poverty is a controversial issue for social scientists, non-governmental organizations, and governments. During the last few decades, research findings have highlighted the extent to which the urban environment influences community life and development. Specifically, the urban environment has been characterized as a strong structural determinant of health that plays an increasingly important role in mental health [4].

Consequently, several studies have underlined the relationship between the physical environment, health, and poverty. The physical environment negatively affects human health in cities, mainly through pollution exposure and constraints on physical activity [5,6]. Environmental pollution negatively influences people’s health conditions, which may cause a decrease in household incomes, leading to a downward spiral [2]. This particular causal association among health and income losses is especially significant in communities with a dearth of goods and public services [7].

Lack of access to basic urban facilities, employment-commercial areas, healthy food, green and leisure areas, and transportation nodes may cause not only physical and mental health deterioration, but also negative economic consequences [8]. Moreover, those living in low-income neighborhoods, especially poor, unplanned settlements, have poorer health outcomes—such as respiratory diseases, obesity, and depression—that may be aggravated by the characteristics of the surrounding urban environment [9,10,11,12].

Although there is substantial literature focusing on the association of community health with poverty and the built environment, there is a general lack of studies assessing the possible direct relationship between the urban environment and socioeconomic outcomes for residents; this is despite the increasing importance that urban dynamics have taken in socioeconomic strategies for cities in practice. Although socioeconomic outcomes include a variety of factors related to wealth, power, and status in society, income is an important measure that is routinely used, when available [13].

Hence, the objective of this work is to investigate the relationships between environmental and urban features and poverty in Detroit, a well-known US city with high poverty rates. We specifically focus on characterizing impacts of environmental and urban risk factors for poverty in the Detroit Metropolitan area by assessing the relationships between changes in household income and characteristics of the built environment. Overall, this work tries to reshape the linear model of these relationships (urban environment affects health which affects socioeconomic outcomes) into a triangular model, characterizing possible direct associations among environmental–urban features and income dynamics.

## 2. Materials and Methods

### 2.1. Study Area and Scope

Detroit, Michigan, the sixth-largest city in the USA, is the case study area for this work (shown in Figure 1a). The city has been an American symbol of urban decay since the late 1960s [14]. It was hit by an economic crisis that turned it into a hub of low salaries, high rates of unemployment, a failed education system, deficient public transportation, an advanced level of abandonment, and lack of investment—a situation described as an urban death spiral [15]. From 2008 to 2010, it was again deeply affected by the auto industry crisis, undergoing a major economic and demographic recession that led to physical and structural urban decline; this further negatively impacted the urbanicity of the city [16]. Over this time, Detroit also experienced significant increases in poverty [17], with the number of high-poverty census tracts tripling from 2000 to 2009; this resulted in the highest poverty rate among large metropolitan areas in the United States [1,18]. Detroit also lost more than 60% of its population, leading to a 22% vacancy rate [19]. In response, the Detroit Strategic Framework Plan was developed, and the city has been designing new strategies to update the Master Plan of Policies in order to transform and revitalize the city [20]. This renewal is currently impacting the metropolitan area through demolition processes located throughout the city.

Historically, Detroit has also suffered from racial segregation and inequality. African Americans and other working-class communities migrated to Detroit during the industrial expansion of the early twentieth century, but were located in segregated neighborhoods and suffered from low wages [15]. During the first economic crisis, many of the poorer Black families stayed in the city while White residents and middle–upper class Black families moved out. In 1980, Detroit had the largest African American population of any city in the country [15]. This racial disparity has continued to today, with 78% of the population identified as African American [21].

Overall, we chose to study Detroit due to its high poverty rates, high level of inequality and segregation, and recent changes to its urban form. Here, we focus on a 5-year time range, from 2013 to 2017, as a post-crisis scenario characterized by the consequences of population loss, unemployment, and urban decay.

### 2.2. Data Sources

To measure the change in socioeconomic outcomes, we obtained data on median household income from the U.S. Census Bureau, American Community Survey [21]. Specifically, we used the 5-year estimates for 2013–2017. Values for all census tracts (*n* = 310) of the Detroit metropolitan area (Detroit city, Highland Park city, and Hamtramck city, see Appendix A) were extracted.

To identify potential environmental and urban features that may influence changes in income, a total of 27 features were considered for analysis. Features were selected based on representing at least one of the following categories of utility or impact: accessibility, environmental pollution, and urban decay. Example features include bus stops, roadways, bike lanes, railways, industry, office—commercial, brownfields, parks, grocery stores, fire stations, libraries, and colleges. Data on the spatial locations of these features were collected from a variety of sources, with the objective of obtaining the most updated information available for the Detroit metropolitan area between 2013 and 2017. A list of all features considered here, along with the sources, time frame, and specifications of data used, is provided in Appendix A.

### 2.3. Analysis Methods

All feature data were georeferenced, mapped, and discretized using geographic information systems software (QGIS.3). To qualitatively discern patterns of location and proximity between urban environmental features and changes in household income, we first compared maps of the native georeferenced data. To enable quantitative proximity comparison, data for all variables were then discretized using a consistent spatial grid, with resolution (cell size) of 1 × 1 mile (see Appendix A), consistent with the dimensions of a walkable neighborhood. The grid orientation was set to align with the northern boundary of Detroit city. Cells of this grid with at least half of their area located within the boundary of metropolitan Detroit—including Highland Park city and Hamtramck city—were used for analysis, resulting in a total of 151 cells.

To measure the change in socioeconomic outcomes for association analysis, we first calculated the difference in median household income (DMHI) for each census tract from 2013 to 2017, such that negative values of DMHI indicate census tracts that experienced a decline in median household income and positive values indicate tracts that increased in income. The difference in median household income for each grid cell (DMHI_cell_) was calculated as the weighted average of values for all census tracts it overlapped. Specifically, we used:DMHI_cell_ = Σ (ρ_i_A_i_ DMHI_i_)/Σ ρ_i_A_i_, (1)
where DMHI_i_ is the difference in median household income (that for 2017 minus that for 2013) in each census tract i, overlapping the cell; ρ_i_ is the population density (population per sq. mile) for census tract i; and A_i_ is the portion of the area (sq. mile) of census tract i that overlaps the cell.

For the remaining features, in order to assess proximity, we used frequency tables (created in QGIS.3) to calculate the occurrence of each environmental and urban variable within each cell. We also calculated the occurrence of each such variable within a buffer area containing the cell and its surrounding eight cells (a total of nine cells of 9 sq. miles). When creating buffer areas, only the 151 previously defined cells were considered. Overall, we used two scales for measuring the frequency of urban and environmental variables, the cell scale—a square of 1 × 1 mile—and its surrounding buffer scale—a concentric square of 3 × 3 miles. (Appendix A provides a map of the cell grid showing each scale.) Specific measures of occurrence (existence, count, length, or area) used for each feature are described in Table 1, Table 2 and Appendix A. All variables considered in this analysis were continuous, except for the binary categorical variable (yes = 1, no = 0) that was used to indicate whether a zone comprises an air quality non-attainment area for sulfur dioxide (NAA-SO_2_). Overall, we calculated values for a total of 27 explanatory variables (measuring environmental and urban factors) and 1 outcome variable (DMHI) for 151 grid cells and 151 buffer areas.

Once values for each variable in each grid cell were obtained, we calculated descriptive statistics for all variables in the population of cells and buffer areas. To measure the direction and strength of association among proximity to urban and environmental features variables and DMHI, we performed correlation analysis (using GraphPad Prism 8 software). Due to the skewness of the data, we calculated non-parametric Spearman correlations.

## 3. Results

Below, we describe the univariate spatial variations found in DMHI and the environmental and urban features in Detroit. This is followed by a description of the bivariate correlations found between DMHI and the environmental and urban features. Finally, we present the correlations found among those potential explanatory variables that had stronger relationships with DMHI.

### 3.1. Spatial Distributions of Each Variable

Figure 1 provides maps of each variable assessed in this study. Descriptive statistics for the gridded values of all variables at both the grid cell and buffer level are provided in Table 1 and Table 2. Distributions of most variables measured at the grid cell level were positively skewed. However, distributions at the buffer level were less skewed and more normal. Variations for each variable are described here.

#### 3.1.1. Change in Household Income

Figure 1b presents the spatial distribution of change in median household income (DMHI) for Detroit between 2017 and 2013 by census tract. A histogram and normal quantile plot of DMHI are provided as Appendix A Appendix A. A total of 41% of census tracts experienced income reduction, with 4.5% of tracts having a household income reduction of more than 10,000 USD. Income loss areas appear evenly scattered throughout the city, with no clear patterns distinguishing the east side and west side of Detroit. However, tracts with the largest reductions are predominantly White or predominantly Black (Figure 1a). In contrast, although tracts with increased household income are also seen throughout metropolitan Detroit, the inner-city area (downtown and its surroundings) have an apparent concentration of increased income, as well as a predominance of mixed racioethnic identity.

The distribution of DMHI at the grid cell level is reasonably normal, with mean of 1791 USD, a standard deviation of 6964 USD, and a range between –13,198 and 47,018 USD. Overall, 11.9% of the cells (18) had minimal household income changes of less than ±1000 USD over the five-year period of analysis, whereas 36.4% had reduced household incomes, with 24, 22, and 9 cells losing 1001 to 3000 USD, 3001 to 6000 USD, and over 6000 USD, respectively. In total, 51.6% had increased incomes, with 27, 26, and 25 cells gaining 1000 to 3000 USD, 3001 to 6000 USD, and more than 6000 USD, respectively; four of the latter cells gained more than 10,000 USD. Overall, there was a substantial tendency toward increasing incomes in Detroit from 2013 to 2017. (Additional descriptive statistics can be found in Table 1.)

#### 3.1.2. Public and Alternative Transportation

Figure 1c maps variables that indicate accessibility and connectivity to public and alternative transportation, including stations for trains or buses, smart bus lines, bike lanes, bus stops, and the airport. Stations for trains and buses are primarily located along the NW–SE and SW–NE axes, with a concentration near downtown and its surrounding. As shown in Table 1 and Table 2, the mean (range) of station count was 0.05 (0–2) and 0.41 (0–4) for grid cells and buffers, respectively. In contrast, smart bus lines have a more radial distribution centered in the downtown area, leaving much of west Detroit poorly connected. The mean (range) of the number of smart bus lines for grid cells and buffers were 1.3 (0–14) and 10.9 (0–67), respectively. Bike lanes are mainly located in the southern area of the city and are especially concentrated in the inner-city. The mean (range) number of lanes (including bike lanes, greenways, and sharrows) was 1.6 (0–13) and 13.3 (0–57) for grid cells and buffers, respectively. For bus stops, a relatively homogeneous distribution can also be seen throughout the city, with concentration along major roadways and a mean (range) of stop count for grid cells and buffers of 35.1 (0–82) and 294 (41–541), respectively. Finally, there is one airport to the east of Detroit.

#### 3.1.3. Urban Facilities

Figure 1d–f displays the geographic distributions of urban facilities and land uses, including governmental facilities (fire stations, police stations), health-related facilities (hospitals, health centers, grocery stores, recreation centers), educational facilities (Head Start children’s centers, public libraries, schools, and colleges), office or commercial areas, parks, and cemeteries. Certain governmental, health-related, and educational facilities, such as fire stations (cell mean count of 0.2, cell range 0–2; buffer mean 2.1, buffer range 0–6), schools (1.5, 0–6; 12.5, 2–31), office or commercial areas (921, 0–3541; 7460, 826–16,662 thousands of sq. feet) and groceries (0.7, 0–7; 6.3, 0–15), are present fairly regularly throughout the city, with some concentration in the inner-city area. Office or commercial activities can also be seen along main transportation routes. However, there appears to be a lower concentration of police stations (0.05, 0–1; 0.4, 0–2) and other educational facilities, including public libraries (0.1, 0–1; 1.1, 0–4), colleges (0.1, 0–2; 0.8, 0–5) and Head Start children’s educational center locations (0.2, 0–2; 1.8, 0–8), in the eastern part of the city. Colleges are mainly concentrated along a downtown–university axis, which also appears to have increased density of many other urban facilities, especially hospitals (0.05, 0–5; 0.4, 0–6) and health centers (0.16, 0–2; 1.25, 0–6). In contrast, recreation centers (0.2, 0–2; 1.9, 0–6), public park areas (628, 0–11.1; 4591, 592–19,397 in thousands of sq. feet), and cemetery areas (660, 0–17,933; 5431, 0–57,662 in thousands of sq. feet) are mainly scattered and distributed far from the center.

#### 3.1.4. Environmental Pollution

Figure 1g presents the geographic distribution of environmental pollution variables, including the locations of the airport, toxic releases, railways, high-volume roadways, industry, brownfields, and an air quality non-attainment area (for SO_2_), with summary statistics in Table 1 and Table 2. Toxic release locations appear to be substantially located in industrial areas, with a mean count of 1.0 (with a range of 0–38) for grid cells and 12.3 (0–194) for buffers. In large industrial areas, the concentration of railways can also be seen, especially in the southwest, with a mean railway length (in thousands of feet) of 3.38 (0–28.4) and 27.5 (0–129) for cells and buffers, respectively. High-volume roadways can be seen throughout the metropolitan area, with higher density apparent near downtown. The mean length (in thousands of feet) of these roadways for a grid cell was 31.3 (0–81.9), and 255 (17.8–51.2) for the surrounding buffer area. Larger industrial areas are located throughout the city, significantly decreasing in the northwest, downtown, and eastern parts of the city. The mean area (in thousands of square feet) is 1332 (with a range of 1.72–9521) and 9897 (36.2–27,760) for grid cells and buffers, respectively. Many brownfield locations can also be seen throughout the city, with a concentration near downtown and its surroundings. The mean (range) of the count of brownfield locations was 4.9 (0–15) and 40.2 (4–89) for cells and buffers, respectively. Finally, the air quality non-attainment area covers the southwestern industrial side of the city.

#### 3.1.5. Urban Decay

Figure 1h presents the geographic distribution of urban decay variables, including locations of demolitions, vacancies, and roadways with poor pavement. The figure shows that during the post-crisis time period studied, the city had a very elevated level of urban decay. Specifically, roads with poor pavement are seen along many main arteries and in the inner-city, with a mean length (in thousands of feet) of 14.0 for cells (with a range of 0.25–48.5) and 116 for buffers (range 13.6–328) for grid cells and buffers, respectively. Vacancy count (cell mean of 311, with a range of 1–966; buffer mean of 2591, with a range of 90–6727) and demolition count (74.9, 0–446; 610, 9–1369) also expansively cover the city, except in areas of industrial land use, and—contrary to the result for poor pavement—in the downtown and its proximity.

### 3.2. Correlations between Income Change and Environmental and Urban Features

Table 3 provides Spearman correlations between change in median household income (DMHI) and each environmental and urban feature variable. Appendix A provides scatter plots for the variables with the highest correlations. The relationships with each variable group are described below.

#### 3.2.1. Public and Alternative Transportation

Comparing Figure 1b to Figure 1c suggests that the cluster of tracts with increasing incomes coincides with the hub of train-bus stations, smart bus lines, and a high concentration of bicycle lanes. However, only weak correlations were found between these features and DMHI. The highest positive correlation was found with bike lanes at the buffer scale (Spearman’s ρ of 0.19).

#### 3.2.2. Urban Facilities

Comparing Figure 1b with Figure 1d–f suggests that areas of high facilities density coincide with the increased incomes hub located in the inner-city. This is supported by the positive correlations found between DMHI and most urban facilities and land-use features. The highest positive correlations were found within the health-related facilities group between DMHI against recreation centers (ρ = 0.23 at the buffer scale, ρ = 0.20 at the cell scale) and groceries (0.20 for the buffer scale). Office and commercial land use had the highest correlation (0.22, 0.19) with DMHI among the variables depicted in Figure 1e. Among educational facilities, the highest positive correlations were found between DMHI and schools (0.19 for the buffer scale) and public libraries (0.18 at the cell scale).

#### 3.2.3. Environmental Pollution

Comparing Figure 1b with Figure 1g, spatial coincidence is not clearly evident between environmental pollution and income losses. Statistical analysis (Table 3) indicates weak positive correlations between DMHI and brownfields (0.24 at the buffer scale, 0.20 at the cell scale), railways (0.20 at the cell level), and industry (0.19 at the buffer level).

#### 3.2.4. Urban Decay

Comparing Figure 1b to Figure 1h suggests that the cluster of tracts with increased incomes coincides with high poor pavement density, reduced vacancy, and limited demolitions. However, vacancies and demolitions show low statistical significance when correlated with DMHI (Table 3); the poor pavement feature stands out with the strongest positive correlation at the grid-cell scale, with Spearman’s ρ = 0.23 for cell data and 0.18 for buffer data.

### 3.3. Correlations among Environmental and Urban Feature Variables

To better understand the potential relationships of the environmental and urban feature variables with household income changes in Detroit, we describe below the relationships among the potential explanatory variables. We focus here on relationships between the variables that have the strongest association with DMHI (as presented in Section 3.2). Appendix A provide detailed lists of the Spearman correlations. The strongest correlations were generally found with the buffer-scale data; hence, values provided here are for the buffer-scale correlations unless otherwise noted.

#### 3.3.1. Public and Alternative Transportation

As shown in Appendix A, correlations of bike lines with other public transportation variables were moderately positive, including for train or bus stations (Spearman ρ = 0.47), smart bus lines (0.46), and bus stops (0.44). Moderate correlations were also found between the number of bike lanes and a few environmental pollution and urban decay features, including for brownfields (0.64), poor pavement (0.59), railways (0.48), and industry (0.46). A weak correlation (0.28) was found for roadways with high traffic. Finally, moderate correlations were found between bike lanes and some urban facility variables, including for fire stations (0.58), health centers (0.47), groceries (0.43), schools (0.38), colleges (0.36), and office or commercial spaces (0.31). Overall, the number of bike lanes (bike lanes, greenways, and sharrows) was moderately correlated with the main public transportation system variables, industrial and polluted area variables, and with several urban facility variables.

#### 3.3.2. Urban Facilities

Within the urban facilities group, school presence was found to be moderately positively correlated with several of the other variables. These include some other educational facilities (public libraries 0.64 and colleges 0.47), health-related features (health centers 0.60, groceries 0.49, and hospitals 0.44), office and commercial areas (0.44), and fire stations (0.41). School presence was also positively correlated with certain environmental pollution and urban decay variables, including brownfields (0.54), poor pavement (0.50), and high-volume roadways (0.44). Lower correlations were found between school presence and the public transportation variables, including bus stops (0.40) and bike lanes (0.38). The presence of public libraries at the cell scale had weak correlations at the same scale with schools (0.34), groceries (0.31), bus stops (0.31), office or commercial areas (0.29), and health centers (0.27), as well as with buffer-scale data on high-volume roadways (0.27). Office and commercial land area were strongly positively correlated with brownfields (0.74), bus stops (0.71), and high-volume roadways (0.71). Moderate positive correlations were found with poor pavement (0.62), public libraries (0.52), recreation centers (0.48), schools (0.44), industry (0.43), colleges (0.43) and groceries (0.43). A few moderate and positive correlations were found between recreation centers and other variables, including office or commercial areas (0.48), groceries (0.41), brownfields (0.41), and the non-attainment area for SO_2_ (0.40). Weaker correlations were found with health centers (0.33), railways (0.33), and high-volume roadways (0.31). Surprisingly, a weak negative correlation was found with the urban decay variables of vacancies (−0.32) and demolitions (−0.28). The buffer scale count of grocery stores was found to be strongly to moderately correlated with the environmental pollution variables (brownfields 0.62, railways 0.49, high-volume roadways 0.44, industry 0.37) and moderately correlated with bike lanes (0.43), bus stops (0.43), libraries and schools (0.46, 0.48), office or commercial spaces (0.43), and recreation centers (0.41).

#### 3.3.3. Environmental Pollution

Brownfields were strongly positively correlated with most of the other environmental pollution variables, including poor pavement (0.73), high-volume roadways (0.68), railways (0.64), and industry (0.62). Brownfields were also found to be strongly to moderately positively correlated to most public transportation variables, including bus stops (0.71), bike lanes (0.64), and train or bus stations (0.43). Brownfields were strongly positively correlated with two of the urban facility variables, specifically office and commercial land use (0.74) and groceries (0.61), and were moderately correlated with schools (0.54), fire stations (0.50), public libraries (0.48), colleges (0.44), police stations (0.42) and recreation centers (0.41). Similarly, industrial land was strongly correlated with railways (0.88), toxic releases (0.70), and brownfields (0.62) and was moderately correlated with office and commercial land use (0.43). Railways (at the cell scale) were mainly associated with the other environmental pollution variables, including with industry (0.67 for the buffer-scale industry data, 0.78 for the cell-scale data), toxic releases (0.50, 0.45), brownfields (0.44, 0.45), and the SO_2_ non-attainment area (0.40 at both scales). Overall, there were large associations among environmental pollution variables and also with other urban factors measured in Detroit. This may be a consequence of the historic hazardous industrial activities that have left their mark on the entire city.

#### 3.3.4. Urban Decay

Poor pavement (at the cell level) was moderately positively correlated with variables from all categories, including bus stops (0.53), bike lanes (0.43), train or bus stations (0.41), brownfields (0.48), high-volume roadways (0.47), colleges (0.43), police stations (0.46), office or commercial areas (0.44), hospitals (0.41), and health centers (0.40). This suggests that urban decay may be present throughout the city.

### 3.4. Summary

We found weak positive correlation between DMHI and many urban factors. Spearman’s rho values were weak overall, but highest for brownfields (ρ_buffer_ = 0.24, ρ_cell_ = 0.20), recreation centers 0.23, 0.20), poor pavement (0.18, 0.23), office and commercial land use (0.22, 0.19), railways (ρ_cell_ = 0.20), groceries (ρ_buffer_ = 0.20), industry (ρ_buffer_ = 0.19), bike lanes (ρ_buffer_ = 0.19), schools (ρ_buffer_ = 0.19), and public libraries (ρ_cell_ = 0.18).

Overall, results suggest that proximity (location within a radius of 1.5 miles) to commercial-office and industrial areas, schools, and public library educational facilities, recreation centers, and groceries was positively associated with increasing household incomes in Detroit during the post-crisis era. However, positive associations were also found between DMHI with brownfields, railways, and poor pavement. This is consistent with the high levels of covariance among the environmental and urban factors studied, particularly between these variables and industrial land uses, which may confound the relationships with DMHI. The rest of the environmental pollution and urban decay features did not have statistically significant associations with DMHI. However, geographic coincidence in the inner-city area of DMHI increases and low rates of vacancy and demolitions are apparent in the maps. Additionally, this study found no associations between green areas and DMHI.

## 4. Discussion

Resilience is defined as the ability to face, cope and adapt positively to external threats with minimal damage or consequences [22]. The concept of resilience can be applied to any system and at different scales [23]. Community resilience is based on an interconnected network of systems—ecological, urban, socioeconomic—that directly impact society [22]. The urban environment may help increase a community’s resilience to disasters. This can occur in two ways: by designing and building cities with fewer vulnerabilities and by responding efficiently to restore previous conditions [24]. This study suggests that certain characteristics of the surrounding urban environment may help the economic resilience of the community.

Detroit is historically characterized by having substantial industrial land use and a vibrant inner city with active urbanicity. The downtown area possesses mixed land uses with a central business district that contains the largest concentration of jobs in the metropolis [25]. Despite being acknowledged as an area with relatively good accessibility, especially by car, people living in the central city are mainly from racial and ethnic minority groups, are less privileged, and are without cars; yet, they must rely on poor public transit services to commute [26,27]. Detroit is also segregated by income and race [28], with urban infrastructure and features differing substantially between neighborhoods.

The original scale of analysis in this study was based on units of 1 × 1 mile in order to represent the influence of walkability on socioeconomic dynamics. Literature shows that the built environment directly affects walking behavior. Parameters such as proximity to destinations, connectivity, density, sidewalks, street connectivity, safety, neighborhood type, and aesthetics encourage people to walk more and farther [29]. Healthy people tend to walk a median of 600 m per journey, although this number increases for those with low incomes to up to 800 m [30]. Yet, low walkability and sedentary behaviors also have been associated with low-income neighborhoods [5], while they usually report more active commuting [11].

The strong increase in poverty during the economic recession in Detroit, in addition to the characteristics of its urban environment, made us hypothesize that Detroit‘s physical and environmental features may have influenced increases in poverty. However, this study highlighted that specific urban features might have helped to promote poverty resilience, specifically the proximity to commercial and office areas, industry, schools, public libraries, recreation centers, groceries, and bike lanes. This suggests that accessibility may be the overarching urban factor that most influences household income improvements. Moreover, downtown Detroit and its surroundings, one of the more mixed racioethnic areas in the city, have an apparent concentration of increased household incomes that coincides geographically with proximity to such facilities or land uses, suggesting that some disadvantaged communities, regardless of race or ethnicity, have seen increases in their household incomes—this is an optimistic outcome against the concentrated and persistent poverty.

Urban accessibility, commonly defined by integrated assessment of transportation and land use to determine how many destinations can be reached per unit time by using a specific mode of transport [31,32], is a key feature in urban inequality studies. It brings out significant differences between poor and high-income neighborhoods regarding the number and types of facilities reached [33], and it is commonly used in food desert studies and employment status assessments [34]. Findings highlight the barriers in impoverished communities to accessing healthy food [35] and the endemic spatial mismatch—or spatial disconnection from job opportunities—that these communities suffer, especially in the US [36].

The proximity to job opportunity areas (office, commercial, and industrial areas here), impacts household economic health [34,36]. Both the statistical correlations and geographic observations of this study support the above relationship, especially for the inner city of Detroit. It suggests that living near job opportunity areas may be an asset for poverty resilience.

The previously discussed poor health–poverty downward spiral [2] provides a framework for understanding the relationship between health and socioeconomic impacts. In this study, proximity to recreation centers and groceries were found to have the highest correlations with household income among the health-related facilities, rather than proximity to health centers and hospitals. This suggests that illness-prevention strategies such as physical activity and healthy food may help to improve household economics. This is consistent with studies indicating that chronic health conditions such as obesity, smoking, mental illness, and musculoskeletal problems are strongly associated with work absence [37,38]. Chronic diseases have more impact on work performance and job instability than acute diseases [39]. They can lead to low salaries, unpaid days off, or even job loss. They are one of the major causes of work incapacity and absenteeism in the world, with an estimated burden of 28.2 million lost workdays annually in the U.S. [40]. Focusing on Detroit, this city shows significantly elevated percentages of disability for every measured group by age or ethnicity [25].

On the other hand, it is necessary to probe the role that proximity to educational facilities may play in socioeconomic outcomes. Results show that proximity to schools, located within a 3 × 3 mile area, may slightly influence income improvements. In contrast, for public libraries, this influence occurs at a smaller scale (1 × 1 mile area). To decipher possible implications, we may focus on working families’ needs. Low wages force low-income households to seek to earn at least two salaries. Therefore, having school-age children stimulates employment-seeking behaviors by working-age family members. Moreover, pre and after-school programs—where students can stay until 6 p.m.—have developed in schools, libraries, and recreation centers to offer safety, security, learning opportunities, and adequate nutrition for children. These have become a necessary and reliable tool for parents, especially for underprivileged populations [41], to support household income improvement. In this sense, proximity to these facilities is very important to disadvantaged communities whose children may travel on their own, either on foot or by bicycle.

No significant results were obtained when assessing relationships between DMHI and green areas, despite the health benefits that proximity to green areas has been found to give [42,43]. This may indicate a low level of usage among the community due to poor access and poor maintenance of the natural spaces in Detroit [44]. This problem has been addressed by the “2017 Parks and Recreation Improvement Plan” of the City of Detroit, rehabilitating parks and creating new green spaces, trails, greenways, and bike paths.

Regarding mobility factors, the only significant correlation found with household income and the public and alternative transportation variables was with bike lanes at the 3 × 3 miles scale. Certain populations—the poor, the elderly, and the youngest—who cannot access auto transportation—often rely on public transportation services for their daily mobility. When public transportation is absent or inefficient, and distances are too great for walking, other means of travel, such as bicycling, must be used. These conditions apply in many urban and suburban areas in the US, where segregated urban planning, added with car-oriented public transportation strategies, worsens urban inequality and constrains the mobility of communities [26,45,46]. In Detroit, which has striking differences between accessibility to auto and public transportation, non-motorized means of transportation may provide an option for disadvantaged residents who do not own a car. Therefore, bicycles may be a key to mobility in a large city with deficient public transportation and a high incidence of low-income populations, such as Detroit.

On the other hand, despite the high occurrence of environmental pollution in this industrial city, we found only weak positive associations between household income and brownfields, industry, and railways. This was unexpected because low-income areas are commonly associated with higher concentrations of air pollutants [5], and environmental justice studies show that poor households are more likely to be located in proximity to sources of air pollution, especially industries and high-density roads [47,48,49,50]. Furthermore, researchers have found evidence of the causal links between urban pollution and chronic diseases such as cancer, cardiovascular, and respiratory problems [51,52]. Therefore, the results found here for environmental pollution may have several interpretations. First, proximity to industry may be representing an urban accessibility feature, indicating that access to job opportunities may be more impactful than potential exposure to pollution. Second, the positive associations with brownfields and railways may be due to the strong interdependency of these variables with urban activity and industrial land use. Additionally, the lack of an inverse relationship between DMHI and pollution features may be due to limitations of the study design for assessing the socioeconomic impacts of pollution. More research is needed using other methods to assess the impact that environmental pollution has on incomes in Detroit.

Urban decay features such as poor basic infrastructure, overcrowding, poor ventilation, sanitation, and poor maintenance have been found previously to have a negative impact on health [53,54]. Furthermore, urban decay, which is endemic in many poor neighborhoods, has been associated with depression and stress [10,55], and is also among the known causes of excess mortality and early health deterioration [56]. Literature also indicates that degraded urban environments can hamper good accessibility and incite crime [57,58]. Empty urban plots are regarded as scars in the city, as unsafe places that can affect a community’s physical and mental health [59]. In contrast, dense cities with good accessibility and low crime attract business and employment, which can improve the economic situation of their inhabitants. In Detroit, large areas of the central city with historic industrial uses were abandoned due to the automobile industry crisis, along with housing, facilities, and commercial buildings. Many of these areas are currently in demolition or renewal processes under the new Detroit Strategic Framework Plan. Nonetheless, we found little association between the urban decay variables and DMHI in Detroit, with only the poor pavement feature showing a weak positive correlation. This may be due to its close dependence on high urban activity. However, there is no statistical evidence that the presence of empty and/or neglected buildings and plots in Detroit negatively impacts nearby household incomes, despite an apparent coincidence in the maps of the inner-city hub of household income improvements and low presence of vacancy and demolitions.

### Limitations

Several limitations apply to this study’s findings. First, we have assumed that the spatial frequency and distribution of the urban indicators studied did not change substantially between 2013 and 2017 (the study period), with data gathering focused on 2015. As shown in Appendix A, most of the urban facility and urban decay data appear to have remained substantially constant in time, but a few environmental pollution variables and public and alternative transportation variables changed more regularly. Second, certain environmental and urban factors are incomplete within the areas of Highland Park city and Hamtramck city because we did not find available data for these cities for all the variables measured. Nonetheless, we retained cells in these areas to allow continuous urban analysis. Hence, results near these areas are less certain. As this is a small portion of the study domain, we do not expect that the overall results were impacted substantially. Additionally, because correlations for the 1 × 1 mile cells were weaker and less statistically insignificant than those for the larger 3 × 3 mile buffers, the discussion of this work is mainly based on data for larger buffer areas. However, this limits us from fully assessing the influence of walking proximity. The biggest constraint on the findings of this study is the large association and co-dependence found among many of the urban and environmental variables. Additionally, the relationship between the built environment and income may be ambi-directional, with the built environment influencing income and vice versa. Due to the complex urban interrelationship and the skewness of the data, our attempts to build effective ordinary or non-linear regression models were not fruitful. Accordingly, we focused on examining both geographic distribution maps and bivariate correlations to explore relationships between urban and environmental features with household income change. Hence, implications that assume causality in either direction should be carefully considered.

## 5. Conclusions

Researchers have drawn attention to factors such as social support, education, and health as strong determinants of family poverty resilience, especially among poor-income populations [60,61,62]. Although it is generally accepted that the built environment has a substantial influence on community well-being, especially that related to social and health issues [4,9,42], less attention has been paid to its interactions with household economics. In this study, we first hypothesized that urban and environmental features would negatively influence the household economy in Detroit. Instead, the results show us the friendly face of the city. We found no significant negative relationships between built-environmental features and income, but instead found several weak positive relationships that support guidelines for the design of poverty resilience in cities. Positive correlations were found between household income change and the proximate presence of certain urban facilities. Specifically, these suggest that job areas (even if polluted), educational assets, and opportunities for well-being near to communities may positively influence household incomes. The presence of nearby local alternative transport may be further supportive of positive income change. Overall, the urban planning ideas suggested here align with current concepts of sustainable and healthy cities, for which mixed uses, good accessibility, efficient public transportation systems, green mobility strategies (non-motorized means of travel, walkability), and quality green areas promote well-being, the local economy, and the social cohesion of inhabitants. Further, they support the conclusion that the design of the urban environment is a useful tool in public health strategies, even in interventions and programs for poverty eradication.

## Figures and Tables

**Figure 1 ijerph-18-06982-f001:**
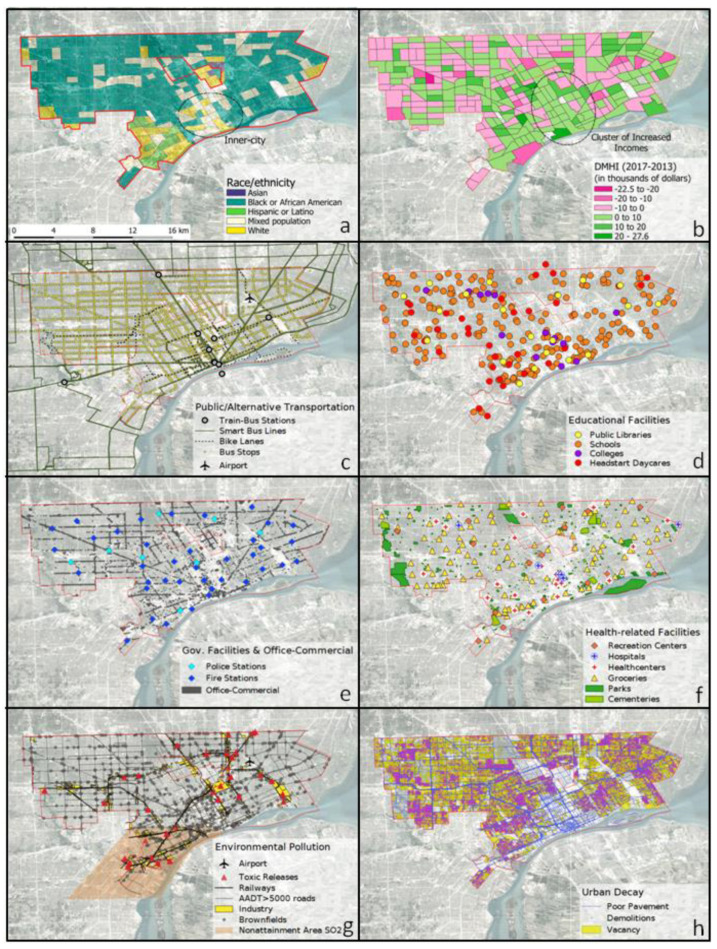
Urban and environmental features of Detroit, Michigan: (**a**) the study area including the city limits (red outline), inner-city location (black oval), and predominant racial/ethnic group by census tract (2011–2015 American Community Survey, U.S. Census Bureau), (**b**) the geographic distribution of difference of median household income (DMHI) between 2013 and 2017 by census tract, (**c**) public and alternative transportation, (**d**) educational facilities, (**e**) government assets and commercial or office areas, and (**f**) health-related facilities and green areas, (**g**) environmental pollution, and (**h**) urban decay features.

**Table 1 ijerph-18-06982-t001:** Descriptive statistics of all variables based on data in each grid cell.

Variable ^a^	Units	N	Mean	Median	Std. Dev.	Minimum	Maximum	Skewness
DMHI 2017–2013 ^b^	US dollars	151	1791	1160	6964	−13,198	47,018	2.31
train/bus stations	count	151	0.05	0.00	0.25	0	2	5.22
smart bus lines	count	151	1.31	0.00	2.22	0	14	3.34
bike lanes	count	151	1.59	0.00	2.48	0	13	2.06
bus stops	count	151	35.16	36.00	16.36	0	82	0.15
airports	count	151	0.03	0.00	0.16	0	1	5.96
toxic releases	count	151	1.04	0.00	3.96	0	38	6.66
railways	feet	151	3377	0.00	5421	0	28,372	2.05
AADT 5000	feet	151	31,281	27,817	15,966	0	81,890	0.73
industry	sq. feet	151	1,332,463	417,467	1,881,540	1722	9,520,648	2.08
brownfields	count	151	4.90	5.00	3.49	0	15	0.63
NAA-SO_2_	yes = 1, no = 0	151	0.09	0.00	0.29	0	1	2.84
poor pavement	feet	151	13,977	12,373	9113	247	48,529	1.15
demolitions	count	151	74.90	62.00	74.48	0	446	1.96
vacancies	count	151	311	298	240	1	966	0.70
public libraries	count	151	0.13	0.00	0.34	0	1	2.19
schools	count	151	1.54	1.00	1.46	0	6	1.07
colleges	count	151	0.09	0.00	0.33	0	2	4.08
Head Start	count	151	0.21	0.00	0.48	0	2	2.27
police stations	count	151	0.05	0.00	0.22	0	1	4.03
fire stations	count	151	0.25	0.00	0.45	0	2	1.38
office/commercial	sq. feet	151	920,798	812,072	599,614	0	3,541,280	1.35
recreation centers	count	151	0.25	0.00	0.46	0	2	1.60
hospitals	count	151	0.05	0.00	0.43	0	5	10.59
health centers	count	151	0.16	0.00	0.45	0	2	2.92
groceries	count	151	0.77	1.00	1.00	0	7	2.33
parks	sq. feet	151	627,966	226,333	1,213,780	0	11,097,581	5.27
cemeteries	sq. feet	151	660,333	0.00	2,523,629	0	17,932,657	4.65

^a^ See Appendix A Appendix A for a description of each variable. ^b^ Difference between the median household income in 2017 minus that in 2013.

**Table 2 ijerph-18-06982-t002:** Descriptive statistics of all potential explanatory variables based on data in the buffer area.

Variable ^a^	Units	N	Mean	Median	Std. Dev.	Minimum	Maximum	Skewness
train/bus stations	count	151	0.41	0.00	0.98	0	4.00	2.74
smart bus lines	count	151	10.93	8.00	11.76	0	67.00	2.33
bike lanes	count	151	13.34	8.00	14.70	0	57.00	1.27
bus stops	count	151	293.70	314.00	110.80	41	541.00	−0.19
airports	count	151	0.24	0.00	0.78	0	4.00	3.64
toxic releases	count	151	12.30	4.00	25.63	0	194.00	5.13
railways	feet	151	27,487	20,413	27683	0	128,924	1.18
AADT 5000	feet	151	255,171	256,643	89,214	17,811	512,574	0.14
industry	sq. feet	151	9,897,417	9,860,497	7,383,379	36,231	27,760,098	0.29
brownfields	count	151	40.18	39.00	19.58	4	89.00	0.34
NAA-SO_2_	yes = 1 no = 0	151	0.78	0.00	2.19	0	9.00	2.88
poor pavement	feet	151	115,604	107,695	59,900	13,561	327,591	1.08
demolitions	count	151	610	636	281	9	1369	−0.14
vacancies	count	151	2591	2478	1505	90	6727	0.46
public libraries	count	151	1.14	1.00	0.98	0	4.00	0.84
schools	count	151	12.53	12.00	5.95	2	31.00	0.82
colleges	count	151	0.77	0.00	1.17	0	5.00	1.71
Head Start	count	151	1.83	2.00	1.74	0	8.00	0.92
police stations	count	151	0.46	0.00	0.53	0	2.00	0.45
fire stations	count	151	2.11	2.00	1.41	0	6.00	0.58
office/commercial	sq. feet	151	7,459,878	7,164,420	2,828,710	826,151	16,662,517	0.33
recreation centers	count	151	1.93	2.00	1.30	0	6.00	0.88
hospitals	count	151	0.46	0.00	1.26	0	6.00	3.36
health centers	count	151	1.25	1.00	1.41	0	6.00	1.45
groceries	count	151	6.34	6.00	2.96	0	15.00	0.17
parks	sq. feet	151	4,590,674	3,814,780	3,355,732	592,477	19,396,548	2.34
cemeteries	sq. feet	151	5,431,453	4058	11,172,928	0	57,662,212	3.06

^a^ See Appendix A Appendix A for a description of each variable.

**Table 3 ijerph-18-06982-t003:** Spearman correlation of the difference in median household income (DMHI) between 2017 and 2013 with potential explanatory variables at the cell and buffer scales.

Variable ^a^	Cell Scale	Buffer Scale
train/bus stations	0.09	0.11
smart bus lines	−0.03	0.10
bike lanes	0.11	**0.19**
bus stops	0.11	0.15
airports	−0.04	0.04
toxic releases	0.03	0.11
railways	**0.20**	0.14
AADT 5000	0.02	0.15
industry	0.07	**0.19**
brownfields	**0.20**	**0.24**
NAA-SO_2_	0.09	0.13
poor pavement	**0.23**	**0.18**
demolitions	0.12	0.14
vacancies	0.12	0.10
public libraries	**0.18**	0.14
schools	0.10	**0.19**
colleges	0.01	0.06
Head Start	0.16	0.15
police stations	−0.02	0.06
fire stations	0.06	0.11
office/commercial	**0.19**	**0.22**
recreation centers	**0.20**	**0.23**
hospitals	−0.09	0.09
health centers	0.11	0.13
groceries	0.03	**0.20**
parks	−0.04	−0.12
cemeteries	0.04	0.02

^a^ See Appendix A Appendix A for a description of each variable. Values in bold had *p*-values ≤ 0.03.

## Data Availability

The main data are contained within the article or Appendix A. Further data of this study are available on request from the corresponding author.

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
