# Peer review of "Healthy Urban Environmental Features for Poverty Resilience: The Case of Detroit, USA"

_ijerph, 2021, doi:10.3390/ijerph18136982_

Round 1

Reviewer 1 Report

A great topic that needs to be researched. The article presents some key insights on the topics of urban poverty and environmental and built infrastructures; however, it has many shortcomings to reveal newer understandings on the relationship and does not offer any breakthrough on the topic. 

  1. The objective was to understand the relationship between urban features and poverty; it would have been informational to know the SES of the 151 census tracts at the household level and correlate with the variables selected for the research. Without this it is hard to understand the relationship just from the median income, which can be affected by many variables.
  2. Detroit has a history of inequality based on race and ethnicity, none of this was discussed in the article (methods section). Any discussion of urban poverty in large cities in the US requires to include this for readers to understand who is affected by poverty. A racial map of the census tracts overlayered with income could be very helpful.
  3. The article needs further description of what buffer and cell scale are for readers who are not fluent with quantitative methods.

Reviewer 2 Report

Referee report for: “Environmental and Urban Features for Poverty Resilience: The Case of Detroit, USA.”

Summary: The paper is a description of how various indicators of social institutions (healthcare, education, job centers, etc.) and of environment influence income outcomes in a city.  The goal is to identify variables important for poverty resilience.  The authors use ACS data at the Census Tract level for Detroit.  The authors use a GIS approach which is compelling and seems to be implemented successfully.

Comments:

  1. The measure of socioeconomic status seems to be median income. This is an economic indicator but doesn’t necessarily capture anything “socioeconomic” (which typically would include some measure of education or occupation or similar).  The paper could be improved by defining the terms from the start as they will be used in the paper and how this links to or deviates from definitions in the literature.
  2. In addition, the authors allude to income “distribution” as opposed to just income levels, but then provide less evidence about the distribution in the statistical analysis. This could be a fruitful area to expand.
  3. As noted by the authors, some of the associations seem nonintuitive and this may be related to various confounders. In addition, the direction of causality may be reverse with incomes affecting the nature of the built environment through mechanisms of taxation and expenditure policies of government.  This would suggest that some of the modeling is in a sense “backwards” in this paper.  The authors do describe in terms of correlations and not causation, which is appropriate, through this feature of the modeling and implications for interpretation could be discussed in more detail especially in the limitations section.
  4. The abstract could be stronger for a public health journal by more clearly indicating the envisioned health link. Similar comment for the title.  The health link seems minimized in the abstract and title but is really the focus of the motivation/intro section so there is a minor disconnect between these pieces of the paper.  The conclusion also suggests that this is less of a health paper and more of a community resilience, regional science themed paper. 

Round 2

Reviewer 2 Report

The authors have addressed my concerns in their revision and I appreciate the new language about causality in the limitations.  Please note that there are remaining editorial corrections necessary in the reference list.  This may be the result of the editorial process as opposed to these authors.  Please check reference 12...author is not "The, L."  Please also note that several of the journal titles are not consistently capitalized.  The reference list needs to be cleaned with an eye to detail before publication.